# The First-Principle Study on Tuning Optical Properties of MA_2_Z_4_ by Cr Replacement of Mo Atoms in MoSi_2_N_4_

**DOI:** 10.3390/nano12162822

**Published:** 2022-08-17

**Authors:** Yongsheng Li, Jiawei Li, Lingyu Wan, Jiayu Li, Hang Qu, Cui Ding, Mingyang Li, Dan Yu, Kaidi Fan, Huilu Yao

**Affiliations:** 1Center on Nano-Energy Research, Guangxi Key Laboratory for Relativistic Astrophysics, School of Physical Science and Technology, Guangxi University, Nanning 530004, China; 2State Key Laboratory of Pollution Control and Resource Reuse, School of the Environment, Nanjing University, Nanjing 210023, China

**Keywords:** two-dimensional material, light absorption, first-principles, ultraviolet (UV)–visible spectra

## Abstract

Recently, with the successful preparation of MoSi_2_N_4,_ an emerging family of two-dimensional (2D) layered materials has been predicted with a general formula of MA_2_Z_4_ (M: an early transition metal, A: Si or Ge and Z: N, P, or As). In terms of this new type of 2D material, how to effectively tune its light absorption properties is unclear. We systematically discuss the effects of replacing Mo with Cr atoms on the lattice structure, energy bands, and light absorption properties of 2D monolayer MoSi_2_N_4_ using density functional theory (DFT) and the Vienna Ab initio Simulation Package (VASP). Additionally, the results show that the single replacement of the atom Cr has no significant effect on the lattice structure of the outermost and sub-outer layers but plays a major role in the accumulation of electrons. In addition, the 2D MoSi_2_N_4_, Mo_0.5_Cr_0.5_Si_2_N_4_, and CrSi_2_N_4_ all have effective electron–hole separation properties. In the visible region, as the excited state increases, the required excitation energy is higher and the corresponding wavelength of light is shorter. It was found that the ultraviolet (UV)–visible spectra are red-shifted when Cr atoms replace Mo atoms in MoSi_2_N_4_; when Cr atoms and Mo atoms coexist, the coupling between Cr atoms and Mo atoms achieves modulation of the ultraviolet (UV)–visible spectra. Finally, we reveal that doping M-site atoms can effectively tune the light absorption properties of MA_2_Z_4_ materials. These results provide a strategy for the design of new 2D materials with high absorption properties.

## 1. Introduction

Since the discovery of single-atomic-layer graphene in 2004 [1], the unique properties of two-dimensional (2D) materials have attracted widespread research interest. The exploration of novel 2D materials has been at the forefront of research on 2D materials. In addition to graphene [2,3], 2D materials, such as 2D boron nitride [4], transition metal sulfides [5], oxides [6], nitride [7,8,9], black phosphorus [10], 2D Mo_2_C [11], WC [12] and 2D TaC [13], have also been developed. Recently, MoSi_2_N_4_ has been successfully prepared [14], as a new member of the 2D materials family. Its unique atomic geometry structure enables it to possess many novel properties. Notably, there is a large family of new 2D materials in the common form of MA_2_Z_4_ (M = Mo, W, V, Nb, Ta, Ti, Zr, Hf or Cr, A = Si or Ge, and Z = N, P or As), including semiconductors, metals and magnetic semiconductors, which will greatly expand the new properties and applications of existing 2D materials.

Based on the first-principles calculations, the mechanical, electrical, magnetic and optical properties of 2D MoSi_2_N_4_ and its family members have been explored [15,16,17,18,19,20,21,22,23,24,25,26,27,28,29,30,31,32,33,34,35,36,37]. For example, Bafekry et al. studied the mechanical properties of monolayer MoSi_2_N_4_. Ghobadi et al. investigated the electronic properties of MoSi_2_N_4_ and WSi_2_N_4_ as channel materials for different field effect transistors [25]. Kang et al. further calculated the second harmonic generation (SHG) of MoSi_2_N_4_. These studies demonstrated that 2D-layered MoSi_2_N_4_ and its family members are promising 2D materials with broad application prospects in various fields, such as sensors, microelectronics and optoelectronic devices [27,28,29,30,31,32]. Recently, their excellent photocatalytic properties have also been preliminarily recognized [33,34] by first-principle calculations. Yu et al. reported novel Janus MoSiGeN and WSiGeN structures with excellent photocatalytic performances [35]. Chen et al. designed and investigated MoSi–N_4_/BlueP heterojunction for overall water splitting [36]. 

Energy bands and light absorption properties play an important role in solar energy, photodetection and photocatalysis applications, etc. There are few reports on the optical absorption properties of MoSi_2_N_4_ materials, and many of the fundamental physical properties and modulation for this class of 2D materials remain unknown. As for the light absorption traits, which atoms play an important role in MA_2_Z_4_ materials? Can the material achieve effective electron–hole separation? Furthermore, is there a strategy to effectively tune the light absorption properties of such materials? In this work, we have analyzed and discussed the stable structures of MoSi_2_N_4_, Mo_0.5_Cr_0.5_Si_2_N_4_ and CrSi_2_N_4_ monolayers through first-principle calculations. The importance of the M-site atoms in the MA_2_Z_4_ material system is demonstrated by analyzing the density of states and energy bands. The excitations of the three materials in the visible region were discussed by time-dependent density functional theory, and the ultraviolet (UV) –visible spectra were simulated. In addition, the trends of the absorption spectra were studied after Mo atoms in2D MoSi_2_N_4_ materials were half and fully replaced by Cr atoms. Based on the analysis results, a strategy to effectively control the light absorption properties of MA_2_Z_4_ materials is proposed. It is expected guide the future research on the MA_2_Z_4_ material family.

## 2. Computation Details

The optimization of the material structure was performed by using density functional theory (DFT) and the Vienna Ab initio Simulation Package (VASP) [38,39]. The PBE (Perdew–Burke–Ernzerhof) method [40] of the generalized-gradient approximation (GGA) calculates the exchange correlation between electrons and nuclei. Additionally, the standard pseudopotential of VASP is adopted as the pseudopotential, which is the result of the interaction of the projection plus the plane wave system. A better accuracy can be achieved through excessive truncation, but it also requires a longer computing time, and with a too high truncation the results can be improved without being more obvious. In order to achieve a reasonable balance between the accuracy and efficiency of the calculations, we choose the cut-off energy of the plane wave basis set to be 500eV. The reciprocal k-points are chosen as 10 × 10 × 1 in structure relaxation and electronic calculation, and the standard for the total energy is 10^−5^ eV. All atomic positions are absolutely relaxed until the force reaches 0.01 eV/Å. A vacuum space of 15 Å along the *z*-axis is used to decouple the potential periodic interactions. It is worth noting that in this study, the effect of magnetism on the material was not considered. In addition, the excited states were estimated by employing the Gauss 09 D01 program combined with time-dependent density functional theory (TDDFT) and using PBE0 exchange-dependent functions and SDD basis sets [41,42] in vacuum. The qvasp [43] and vaspkit [44] programs and the Multiwfn 3.8(dev) code [45,46] also contributed to this work.

## 3. Results, Analysis and Discussion

Figure 1 shows the top and side views of MoSi_2_N_4_, Mo_0.5_Cr_0.5_Si_2_N_4_, and CrSi_2_N_4_ supercells, consisting of 3 × 3 simple cells, respectively, and the data show that all three materials have the P-6M2 (D3H-1) space point group. Among them, the monolayer MoSi_2_N_4_ has a lattice constant of 2.91 Å (Figure 1a) and is formed by a 7-layer atomic stacking of N–Si–N–Mo–N–Si–N, with the two sides having a central Mo atomic layer, forming a symmetric structure, which can be considered a unique structure of this 2D material. In order to analyze the stability of the Mo–N and the Si–N layers, the distances between the first layer and the second layer, the second layer and the third layer, and the third layer and the fourth layer are defined as H_1_ (0.51 Å), H_2_ (1.75 Å,) and H_3_ (1.25 Å), respectively, which indicate strong chemical interactions between the 2D MoSi_2_N_4_ layers. Furthermore, the front and left views of the half- and full-replacement of Mo atoms by Cr atoms in the monolayer MoSi_2_N_4_ are also plotted in Figure 1b,c. It is clear that Mo_0.5_Cr_0.5_Si_2_N_4_ and CrSi_2_N_4_ have similar geometries but differ in that their lattice constants are 2.91 Å and 2.92 Å, and the values of H_1_, H_2_ and H_3_ are (0.56, 1.75, 1.18) Å and (0.51, 1.75 and 1.14) Å. This indicates that the Si–N bonds in the second and third layers do not change (both 1.75 Å) as Mo atoms are replaced by Cr atoms, while the Mo–N bonds in the third and innermost layers gradually decrease. The reason for this is the stronger attraction between the N and Cr atoms in the third layer and the higher overlap of the electron cloud. It is worth mentioning that the distances between the outermost and second layers of MoSi_2_N_4_ and CrSi_2_N_4_ are both 0.51 Å. However, the N–Si bond length of Mo_0.5_Cr_0.5_Si_2_N_4_ significantly increases to 0.56 Å. This may be because the interaction between Mo and Cr atoms in the innermost layer of the structure affects the bond energy between the outermost and the second layers of N–Si. 

Then, the density of states, orbital information and energy band structures of 2D MoSi_2_N_4_, Mo_0.5_Cr_0.5_Si_2_N_4_ and CrSi_2_N_4_ are plotted in Figure 2. As presented in Figure 2a_2_, the band gap of MoSi_2_N_4_ is 1.74 eV, which is consistent with the results reported in previous literature [47,48,49]. The fact that the PDOS of Mo, Si and internal N atoms are all near the Fermi level suggests that all kinds of atoms are involved in the contribution. Mo atoms make most of the orbital contribution during excitation. It is obvious that the Mo atom plays a decisive role at the top of the valence band (VB) and at the bottom of the conduction band (CB). To further elucidate the contribution of the Mo atom, the inset in Figure 2a_1_ shows that near the Fermi level, the s and p orbitals of the Mo atom contribute very little and are almost entirely generated by its d orbital electrons. Furthermore, the energy band diagram displays that the 2D MoSi_2_N_4_ has an indirect band gap with the highest occupied molecular orbital (HOMO) at the Γ point and the lowest unoccupied molecular orbital (LUMO) at the M point.

Figure 2a shows that MoSi2N4 has an indirect bandgap of 1.74 eV, and the band gap of Mo_0.5_Cr_0.5_Si_2_N_4_ is much smaller than that of MoSi_2_N_4_, only 0.81ev (as shown in Figure 2b), The band gap of CrSi_2_N_4_ is even smaller, only 0.27eV (as shown in Figure 2c). A possible reason is that the spatial distribution of the 3d orbitals of the Cr atoms is smaller than that of the 4d orbitals of Mo. Unfortunately, both Mo_0.5_Cr_0.5_Si_2_N_4_ and CrSi_2_N_4_ have indirect band gaps, where the minimum conduction band (CBM) and the maximum valence band (VBM) of Mo_0.5_Cr_0.5_Si_2_N_4_ are located at the S and Γ points, respectively, While the VBM and CBM of CrSi_2_N_4_ are at Γ and M points, respectively. In order to clarify the absolute band positions of the materials, the CBM and VBM of 2D MoSi_2_N_4_, Mo_0.5_Cr_0.5_Si_2_N_4_ and CrSi_2_N_4_, which are (−2.45 eV and −0.71 eV), (−2.74 eV and −1.93 eV) and (−2.79 eV and −2.53 eV), respectively, were obtained. The comparison shows that replacing the Mo atoms in MoSi_2_N_4_ with Cr reduces the band gap and shifts its position to negative energy values. According to the limited analysis of these MA_2_Z_4_ materials, the more active the valence electrons of the M atoms are, the larger the band gap may be.

It is well known that the effective electron–hole separation is an important property of catalytic materials. To verify this, the electron density of individual atoms was subtracted from the molecular charge density after structural optimization, and the bonding charge density difference maps of MoSi_2_N_4_, Mo_0.5_Cr_0.5_Si_2_N_4_ and CrSi_2_N_4_ were obtained, as shown in Figure 3.

Figure 3a displays that the charge accumulation is mainly near the N atoms close to Mo, while the holes are mainly near the Mo atoms, which also fully indicates the existence of effective electron–hole separation. For the MoSi_2_N_4_, Mo_0.5_Cr_0.5_Si_2_N_4_ and CrSi_2_N_4_, it is obvious that the number of electrons flowing from the outermost N atomic layer to other N atomic layers on both sides of the central layer is dominant, while there are also a small number of electrons on the Si layer and the central layer directed to the N atomic layers on both sides of the central layer atomic layer. The charge distributions observed in 2D Mo_0.5_Cr_0.5_Si_2_N_4_ and CrSi_2_N_4_ (Figure 3b,c) are similar to that of MoSi_2_N_4_. However, it can be seen from the figure that there is a difference in the charge distribution between the Mo–N and Cr–N bonds. For example, the yellow hole cloud around the N atom in the Mo–N bond is larger and not spherical compared with that of the Cr atom in the Cr–N bond. The reason for this phenomenon may be that the radius of Mo atom is larger than that of Cr atom, and the valence electrons receive weaker binding forces. In conclusion, the bonding charge density difference maps show that MoSi_2_N_4_, Mo_0.5_Cr_0.5_Si_2_N_4_ and CrSi_2_N_4_ have effective electron–hole separation properties and meet the requirements for use as catalytic materials. 

Finally, to explore the optical properties of MoSi_2_N_4_, Mo_0.5_Cr_0.5_Si_2_N_4_ and CrSi_2_N_4_, the first 100 excited states of these three materials were calculated using (time-inclusive density functional theory) TDDFT at the PBE0/SDD level, which is plotted in Figure 4. The black dashed lines are used to determine excited states with oscillator intensities greater than 0.01, while the dark red and blue dashed lines are used for the visible region (380–760 nm) of interest. It is not difficult to uncover that the monolayer MoSi_2_N_4_ has three strong oscillator positions in the visible light region, corresponding to the excitations from S0 toS19, S0 to S54 and S0 to S63, respectively; the monolayer Mo_0.5_Cr_0.5_Si_2_N_4_ and monolayer CrSi_2_N_4_ have 4,3 strong oscillators positions corresponding to (S0 to S28, S0 to S32, S0 to S46, S0 to S57) and (S0 to S37, S0 to S40, S0 to S88). The relevant excitation energies, the corresponding optical wavelengths and the contributions of the relevant orbitals to the excited states (the contribution value is greater than 5%) are listed in Table 1. It is demonstrated that for the monolayer MoSi_2_N_4_, as the number of excited states increases, the required excitation energy will be higher and the wavelength of the absorbed light will be shorter. For each excited state in the visible range, the excitation is contributed by multiple molecular orbital transitions rather than being dominated by a single molecular orbital. This reflects the complexity of electron transfer at each orbital during excitation. For example, the excitation of S0→S19 has two major orbital contributions: the electrons transfer from the HOMO-1 orbital to the LUMO + 2 orbital contributes about 74.4% to this excitation, while the electrons transfer from the HOMO to the LUMO + 2 orbital contributes about 18.1%. Similar to the monolayer MoSi_2_N_4_, each excited state of the Mo_0.5_Cr_0.5_Si_2_N_4_ and CrSi_2_N_4_ monolayers has a complex orbital contribution. In summary, for any of them, the required excitation energy is positively correlated with the number of excited states, while the wavelength of the corresponding absorbed light has a negative correlation with the number of excited states; in the MoSi_2_N_4_, Mo_0.5_Cr_0.5_Si_2_N_4_ and CrSi_2_N_4_ monolayers, none of the excited states has a decisive orbital transition (contribution greater than 85%), which reflects the complexity of the excitation process.

Figure 5 shows the simulated ultraviolet (UV)–visible spectra of MoSi_2_N_4_, Mo_0.5_Cr_0.5_Si_2_N_4_ and CrSi_2_N_4_ using a Gaussian spread function with a full width half maximum (FWHM) of 0.4 eV. MoSi_2_N_4_ has a wider visible light absorption range (as shown in Figure 5a) with several strong absorption bands located in the ultraviolet range at 353.7 nm, the purple range at 424.5 nm (purple), the green range at 518.3 nm (green) and the infrared range at 764.7 nm. The visible region is consistent with the experimental values at 330 nm and 530 nm, Figure 5b demonstrates that the monolayer Mo_0.5_Cr_0.5_Si_2_N_4_ has two absorption peaks near 400 nm and 600 nm, mainly corresponding to the excitations from S0 to S28 and S0 to S57.

In comparison, strong absorption bands of CrSi_2_N_4_ are located at 450.3 nm (blue) in the blue range and 614.8 nm (red) in the red range (Figure 5c). The peak at 614.8 nm is mainly from the degenerate S0→S37 with an oscillator strength of 0.023, the only oscillator with strength above 0.02. The ultraviolet (UV)–visible spectra of the three materials are plotted in Figure 5d. Surprisingly, the substitution of Cr atoms in the monolayer MoSi_2_N_4_ can slightly reduce the absorption peak heights of the UV and visible absorption spectra, leading to a redshift in the spectra. The major reason for this phenomenon may be that the monolayer CrSi_2_N_4_ is more likely to absorb low-energy photons (energy). More interestingly, after the Mo atoms in the monolayer MoSi_2_N_4_ are half-replaced by Cr atoms, the ultraviolet (UV)–visible absorption spectra split from the previous single peak at 424.5 nm to double peaks near 400 nm and 600 nm, and the positions of these two peaks do not overlap with the monolayer CrSi_2_N_4_ as well as the monolayer MoSi_2_N_4_. This indicates that monolayer Mo_0.5_Cr_0.5_Si_2_N_4_ not only has some properties of monolayer MoSi_2_N_4_ and monolayer CrSi_2_N_4_ but also has the coupling effect between Cr atoms and Mo atoms. Most importantly, this proves that the light absorption properties of such materials can be tuned by central atom doping. This will facilitate property prediction of MA_2_Z_4_ materials in the future and provide an effective method to tune their properties.

## 4. Conclusions

Based on the first-principle calculations, replacing Mo with Cr atoms in monolayer MoSi_2_N_4_ the lattice structure, energy bands, and light absorption properties has been investigated systematically. It was found that the Mo–N bond length in MoSi_2_N_4_ is shortened when Cr is doped at M site, but the overall lattice structure changes little. The doping of Cr narrows the band gap and modulates the energy band structure. Three types of 2D materials of MoSi_2_N_4_, Mo_0.5_Cr_0.5_Si_2_N_4_ and CrSi_2_N_4_ have the bandgaps of 1.74 eV, 0.87 eV and 0.21 eV, respectively. MoSi_2_N_4_ is the direct bandgap material, but Mo_0.5_Cr_0.5_Si_2_N_4_ and CrSi_2_N_4_ are the indirect bandgap materials. The M-site atoms are the main contributors to the Fermi level. The further analysis shows that there exists effective electron–hole separation where the electrons are mainly close to the N atoms while the holes are mainly near the Mo atoms. Finally, through the ultraviolet (UV)–visible spectra, we found that after Mo atoms are replaced by Cr atoms, the absorption peak is red-shifted and the intensity is reduced. When Mo atoms in the MoSi_2_N_4_ are replaced by half of Cr atoms, its light absorption properties in the visible region are tuned due to the coupling of Cr and Mo atoms. This provides a method for modulating the light absorption properties of MA_2_Z_4_ materials and designing a material with high-performance of absorption.

## Figures and Tables

**Figure 1 nanomaterials-12-02822-f001:**
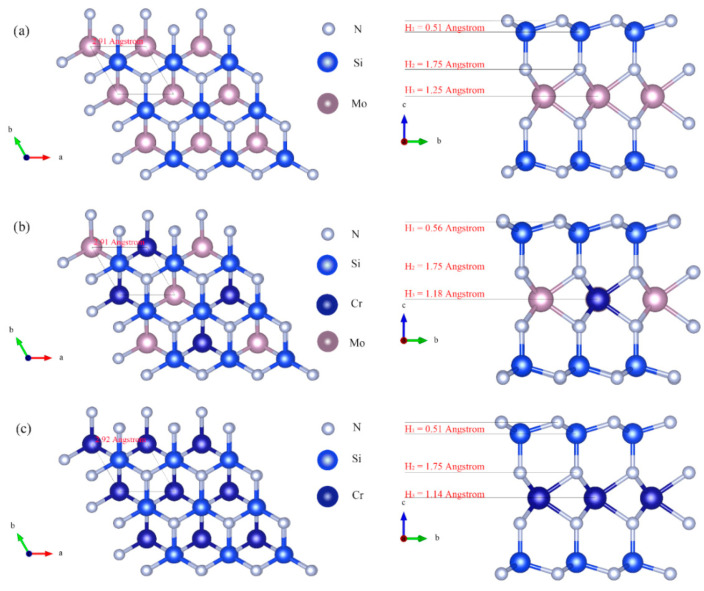
Top (up) and Side (down) views of the optimized geometry for (**a**) MoSi_2_N_4_, (**b**) Mo_0.5_Cr_0.5_Si_2_N_4_ and (**c**) CrSi_2_N_4_ monolayers. Gray, blue, light magenta and navy blue represent Si, N, Mo and Cr atoms, respectively. The black diamonds represent the unit cells in our calculations.

**Figure 2 nanomaterials-12-02822-f002:**
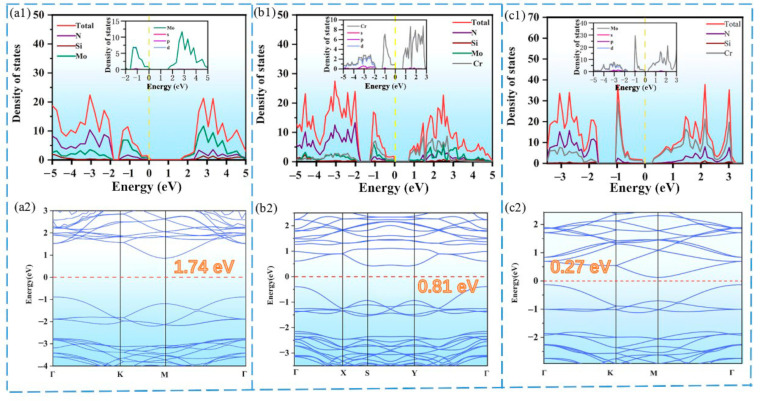
Total density of states (DOS), partial DOS (PDOS) in 2D structures and band of (**a****_1_**) MoSi_2_N_4_, (**b_1_**) Mo_0.5_Cr_0.5_Si_2_N_4_ and (**c_1_**) CrSi_2_N_4_, in which the PDOS of orbitals of the corresponding middle layers are also shown as an insert. The band structure of each type of atom is shown in Figure 2 ((**a_2_**) MoSi_2_N_4_, (**b_2_**) Mo_0.5_Cr_0.5_Si_2_N_4_ and (**c_2_**) CrSi_2_N_4_), which taking the symmetry point of the Brillouin zone as the *X*-axis, where K, M, etc., are the high symmetry points of the Brillouin zone of the crystal, which vary according to the symmetry of the point group of the crystal.

**Figure 3 nanomaterials-12-02822-f003:**
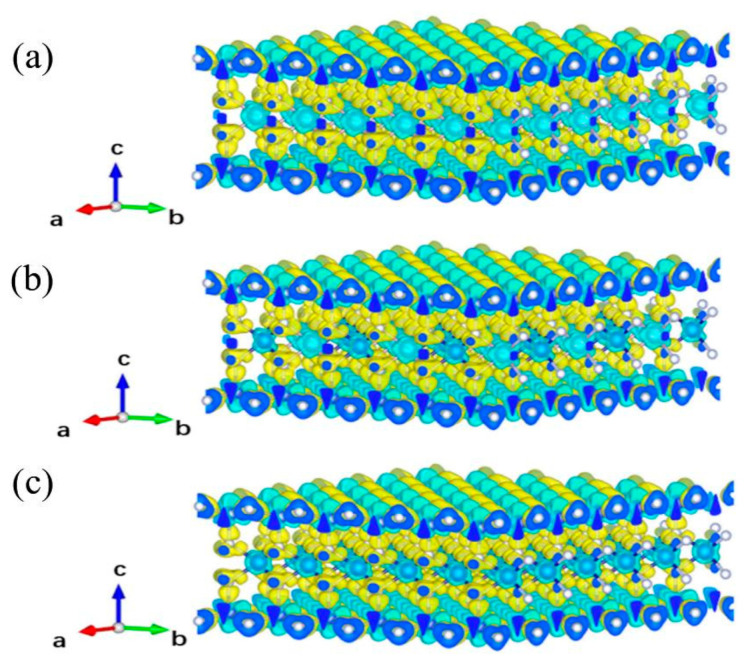
3D charge density difference in (**a**) MoSi_2_N_4_, (**b**) Mo_0.5_Cr_0.5_Si_2_N_4_, and (**c**) CrSi_2_N_4_ with an isosurface of 0.015 e/e/Å3. Yellow and blue isosurfaces represent charge accumulation and depletion, respectively.

**Figure 4 nanomaterials-12-02822-f004:**
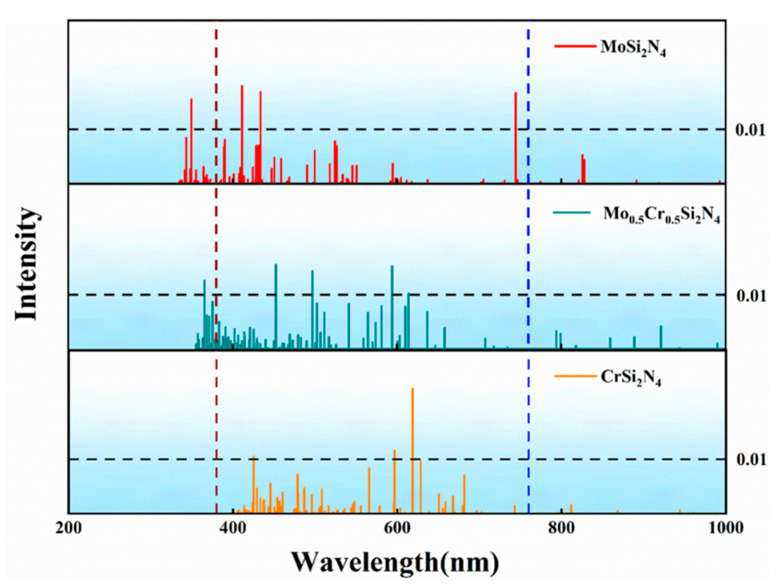
The oscillator strength of MoSi_2_N_4_, Mo_0.5_Cr_0.5_Si_2_N_4_ and CrSi_2_N_4_.

**Figure 5 nanomaterials-12-02822-f005:**
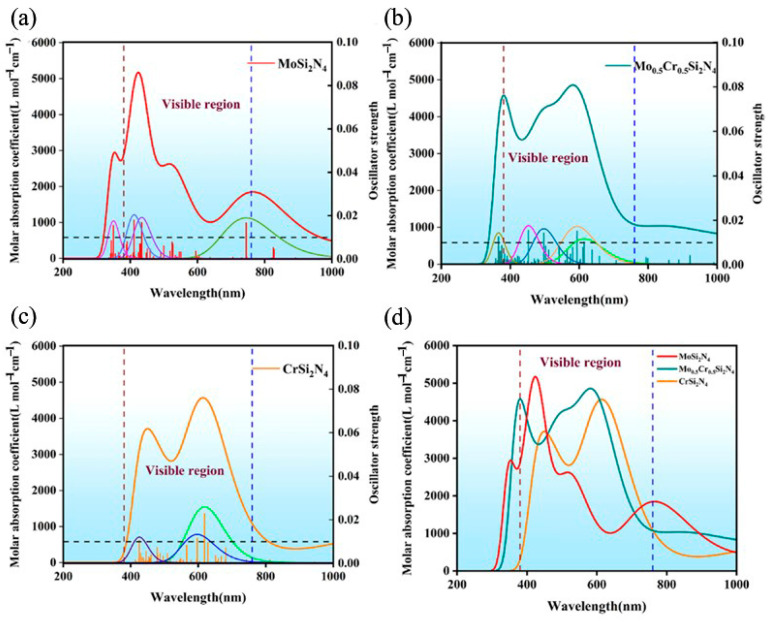
(**a**) The ultraviolet (UV)visible spectra of MoSi_2_N_4_ (red); (**b**) The ultraviolet (UV)visible spectra of Mo_0.5_Cr_0.5_Si_2_N_4_ (dark cyan); (**c**) The ultraviolet (UV)visible spectra of CrSi_2_N_4_ (orange); The peaks of other colors represent various excited states whose contribution intensity is greater than 0.01. (**d**) The ultraviolet (UV)–visible spectra of MoSi_2_N_4_ (red), Mo_0.5_Cr_0.5_Si_2_N_4_ (dark cyan) and CrSi_2_N_4_ (orange).

**Table 1 nanomaterials-12-02822-t001:** The excited state, the excitation energy, corresponding light wavelength and orbital contribution of 2D MoSi_2_N_4_, Mo_0.5_Cr_0.5_Si_2_N_4_ and CrSi_2_N_4._ L: LUMO (Lowest Unoccupied Molecular Orbital); H: HOMO (Highest Occupied Molecular Orbital).

Material	MoSi_2_N_4_			Mo_0.5_Cr_0.5_Si_2_N_4_				CrSi_2_N_4_		
Excited state	#19	#54	#63	#28	#32	#46	#57	#37	#40	#88
The excitation energy (eV)	1.6650	2.8571	3.0137	2.0184	2.0876	2.4937	2.7388	2.0038	2.0775	2.9155
Corresponding wavelength(n m)	744.65	433.95	411.40	614.27	593.91	497.19	452.70	618.75	596.80	425.26
Orbital contribution	H − 1 > L + 2 74.4% H > L + 2 18.1%	H − 3 > L + 4 51.8% H − 8 > L + 2 16.6% H − 6 > L + 2 16.3%	H − 17 > L + 1 42.6% H − 13 > L + 1 28.9%	H − 3 > L + 2 24.8% H − 2 > L + 2 15.2% H − 12 > L 11.9% H − 6 > L + 1 11.7% H − 4 > L + 2 10.3%, H − 7 > L 5.5%	H − 10 > L 19.8% H − 13 > L 17.8% H − 12 > L 16.0% H − 4 > L + 2 10.0% H − 10 > L + 1 5.2%	H − 1 > L + 4 19.3% H − 15 > L 18.0% H − 1 > L + 6 9.2% H − 1 > L + 5 6.3% H − 2 > L + 3 5.5%	H − 1 > L + 6 43.5% H − 1 > L + 4 13.2% H − 1 > L + 8 7.8% H − 17 > L 7.4%	H − 4 > L + 2 38.9% H − 5 > L + 1 12.9% H > L + 6 9.0% H − 4 > L + 4 8.4%	H − 1 > L + 7 35.1% H − 5 > L + 2 9.0% H > L + 8 8.7% H − 2 > L + 2 7.5% H − 2 > L + 8 7.3%	H − 18 > L 21.9% H − 22 > L 14.4% H − 15 > L 11.2% H − 3 > L + 7 5.2% H − 27 > L 5.1%

## Data Availability

Not applicable.

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
