# Peer review of "The First-Principle Study on Tuning Optical Properties of MA2Z4 by Cr Replacement of Mo Atoms in MoSi2N4"

_nanomaterials, 2022, doi:10.3390/nano12162822_

Round 1
Reviewer 1 Report
This ms may be interesting to the audience of the journal. However, it has several issues and I do not recommend it in its current form:
1- Several parts of the paper is not understandable. For instance, the first line of the abstract contains a term "MA2Z4", which is not defined. The paper suffers from significant writing issues that must be corrected talking helps from any native writer.
2- The abstract does not explicitly clarify what kind of "First Principles Calculations" was undertaken?
3 - For example, what atoms play an important role in MA2Z4 materials? Important role in what? What specific important role? What specific property? What specific feature?
4 - Can the material achieve effective separation of electron holes? So, where and why do the electrons and holes need a separation? what for?
5 - The PBE level of theory is a poor selection by the authors. Why did not use meta-GGA SCAN and HSE06 ?
6 - Section 3 is "3. Conclusion and Analysis" - It should be "Results, analysis and discussion". The conclusion section should be separated out.
7- Geometric details of crystals computed are neither presented, not discussed.
8 - In the band structure (Fig. 2, bottom), labeling of X-axis is missing.
9 - Charge density difference plot is noisy. It does not give any information.
10 - Table 1 is very noisy. The meaning of the letters such as L and H, etc, is not described.
11- Analysis and conclusion of the paper is poorly presented.
12 - Background references are not up to date, and the number is low.
Reviewer 2 Report
The manuscript by Yongsheng Li et al., The first-principle study on tuning optical properties of MA2Z4 by Cr replacement of Mo atoms in MoSi2N4. This family of the 2D compounds can be used for design of new 2D materials with high absorption properties by doping M-site atoms of the MA2Z4 compounds.
The manuscript includes an overview of the previous studies published in the literature. This family of 2D compounds has been discovered recently. For this reason, new ways to design functional materials based on this family is original and deserves publication. VASP calculations are done for the doped and pristine MoSi2N4 showing the ways to vary the band gap from 0.81 to 1.74 eV. The results demonstrate the way to substantially change optical properties of MoSi2N4 in order to tune these properties as can be checked from the calculated UV-Vis spectra. The conclusions of the work are supported by the results for the Cr doping. The manuscript is well-written. It can be published in Nanomaterials as it is.
Reviewer 3 Report
This is an interesting study of transition metal alloying in MoSi2N4. The authors find significant changes in bonding and optical properties with Cr substitution for Mo. The manuscript is in principle suitable for publication, but has some flaws that should be addressed before that.
1. In the introduction theory and experiment are mixed together. I think that it would be better if the authors clearly summarize what the current experimental situation and knowledge are and then discuss theoretical works that suggest various interesting properties.
2. The methods section presents the computational parameters. The authors should add discussion of how these parameters such as planewave cutoff where decided on and any tests that were done.
3. I would recommend using the language "electron-hole separation" instead of "separation of electron holes", which is not very clear.
4. I am not very clear on the discussion of catalytic activity based on the orbital characters. This seems to be a very weak argument, especially considering the small band gaps which would not generally point to photocatalytic behavior. They should carefully consider this discussion and modify it to state the issues and what the problems might be.
5. The discussion of bonding is not as clear as it could be. For example, they discuss a lengthening of the Si-N bond as a function of Cr alloying. Is this because of a bond competition between the Si-N bond and the Cr-Si or Cr-N bonds This should be discussed. The comment about Cr being smaller than Mo seems a bit trivial.
6. I do not understand the point of having the sentence "It is showed that for the MoSi2N4 monolayer, as the number of excited states increases, the required excitation energy will be higher and the wavelength of the absorbed light will be shorter." They should reconsider this.
7. In the conclusions they discuss the "UV-vis" optical properties, but with the band gaps and absorption spectra that they find "UV" does not seem very relevant.
Round 2
Reviewer 1 Report
The authors of this work have attempted to answer my questions. They have also revised his paper. However, I am not satisfied with the quality of the presentation and the writing style. While the basic scientific considerations may be valid, I find it difficult to read through this paper. The paper is littered with many misleading phrases. The word "orbital" has shown as "orbit," for example. I have never seen such a term as "highest occupied orbit" in the scientific literature. As I said, there are serious problems with the writing of this paper, and I do not recommend that this work be published in its present form. The reason for this is obviously the poor quality of the English language used, starting from the first line of the abstract. I strongly recommend that the author improve the quality of the paper with the help of a Native English writer. Once the writing problem is resolved, I would be happy to read the paper again.
Reviewer 3 Report
I think that this is now suitable for publication.
Round 3
Reviewer 1 Report
Authors of this work have revised the paper and I see some reasonable improvement in terms of presentation of results. However, I do not agree with the results that were obtained using method PBE, which is surely is not current state of art. To confirm the band gap features as presented in Fig. 2, I ask the authors compute the bandgap of the presented materials using HSE06, in which case, PBE lattice geometries may be used. This is surely doable with any minimal workstation with a 32-64 cores!!! Otherwise, I cannot rely on the PBE-based results discussed in this article.
